# Comparison of the Usability of Eye Drop Aids and the Conventional Bottle

**DOI:** 10.3390/jcm10235658

**Published:** 2021-11-30

**Authors:** Gali Brand, Idan Hecht, Zvia Burgansky-Eliash, Liron Naftali Ben Haim, Duncan Leadbetter, Oriel Spierer, Asaf Achiron

**Affiliations:** 1Department of Ophthalmology, Edith Wolfson Medical Center, Holon 5822012, Israel; galibrand21@gmail.com (G.B.); idanhe@gmail.com (I.H.); lironbh4690@gmail.com (L.N.B.H.); achironasaf@gmail.com (A.A.); 2Sackler School of Medicine, Tel Aviv University, Tel Aviv 6997801, Israel; zviaeb@gmail.com; 3Department of Ophthalmology, Meir Medical Center, Kfar Saba 4428164, Israel; 4Department of Ophthalmology, Bristol Eye Hospital, University Hospitals Bristol NHS Foundation Trust, Bristol BS1 2LX, UK; dcsleadbetter@gmail.com

**Keywords:** eye drops, eye drops administration aids, bottle tip contamination, compliance to treatment, glaucoma treatment

## Abstract

(1) Background: Eye drops are the most common route of administration for ophthalmic medications. Administering drops can be a major hurdle for patients, potentially resulting in noncompliance and treatment failure. The purpose of this study is to compare the efficacy and safety of two different aids and the conventional bottle for eye drop instillation; (2) Methods: An interventional crossover study involving standard eye drop bottle, Opticare aid and Autodrop aid. The study included healthy subjects without a history of regular eye drop use; (3) Results: Twenty-six subjects were enrolled. Of those subjects, 96% and 92% were able to assemble the eye drop bottle into the Autodrop and the Opticare aids, respectively. Subjective assessment indicated that Autodrop was significantly easier to assemble than Opticare (95% CI: −1.6802 to −0.1659, *p* = 0.02). When using either aid, there was no contamination of the bottle tip, which occurred in 46% of subjects when no aid was used (*p* = 0.0005). Fewer drops were expelled when using the conventional bottle as compared to the aids (*p* = 0.05 compared to Autodrop, *p* = 0.1 compared to Opticare); (4) Conclusions: Autodrop and Opticare can assist patients with eye drop placement. These aids completely prevented bottle tip contamination, which was frequently observed when the conventional bottle was used alone.

## 1. Introduction

Eye drops are by far the most common route of administration for medications to treat ophthalmic diseases. Most drops are supplied in a small plastic bottle that is designed for direct instillation into the eye.

The ability of patients to use eye drops independently and successfully can present a major challenge, potentially leading to noncompliance and treatment failure [1,2,3]. Noncompliance may occur due to difficulty with accurate eye drop instillation or fear of self-injury during attempted administration [2]. When asked, as many as 25% of patients report missing doses due to these obstacles [4]. Successful eye drop instillation is an acquired skill, and perfecting the technique requires practice. In addition, bottle tip related traumatic injury and contamination need to be avoided [5,6]. Ophthalmic diseases are more prevalent in the elderly population, where additional comorbidities such as hand tremor, arthritis, poor coordination and peripheral neuropathy present additional challenges to successful eye drop administration [2,7]. In some ophthalmic pathologies, poor adherence to treatment can lead to disease progression and visual deterioration, potentially warranting invasive treatments such as periocular injections or surgery [8], with the associated economic costs [9]. Simply, if the complexity of treatment is resolved, compliance and safety of treatment can be improved [2].

Different aids to administer eyedrops, aimed to help patients to self-administer drops, are readily available for purchase [5,10,11,12,13]. Several studies have demonstrated their effectiveness as compared to controls [4,5,10,11,14,15]. Studies have shown that the pressure required to expel eyedrops with an aid is easier to achieve than the “tip pinch” needed for conventional eye drop bottles [16,17]. This can be significant in patients with motor difficulties, such as in rheumatoid arthritis and Sjögren’s syndrome [15]. In addition, 13% of patients who needed the help of their spouse in eye drop instillation became independent with the use of an aid [15]. Regarding bottle tip contamination, studies have shown that less than half of patients with glaucoma or other ocular diseases were able to instill eye drops without the tip of the bottle touching their eyes [10,18]. Nevertheless, whether these aids can help is not clear. It was reported that using the Opticare aid reduced difficulty controlling the number of drops [15]. However, when using the Xal-Ease, the number of eye drops dispensed was significantly higher than without the device [11].

The Autodrop (Owen Mumford Ltd, Woodstock, England) and Opticare (Cameron Graham Limited, Huddersfield, England) devices are two of the aids commercially available (Figure 1). Both are compatible with most eye drop bottles. When using the Autodrop, the eye drop bottle clips into place, while the body of the device prevents blinking by keeping the lower eyelid open. It has a small pinhole that directs eyesight upward and away from the descending drops. Patients apply pressure directly onto the bottle to expel eye drops. When using the Opticare device, the eye drop bottle is placed inside the device, enabling a better grip. The device is placed on the eye and the eyepiece holds the upper lid, which helps overcome the blinking reflex while administering eye drops.

To the best of our knowledge, no study that directly compared the efficacy and safety of different aids has been published [19]. Therefore, in this study, we sought to compare the efficacy and safety of two different aids and the traditional eye drop bottle.

## 2. Materials and Methods

### 2.1. Study Design/Procedures

This was an interventional, randomized, crossover study. Initially, multiple readily available dispensing aids were evaluated by the local institutional review board committee, including by the hospital’s medical engineer and an infectious disease specialist. These aids included the Opticare, the Opticare Arthro 5, the Opticare Arthro 10 (manufacturer: Cameron Graham Limited, Huddersfield, England) and the Autodrop (manufacturer: Owen Mumford Ltd., Woodstock, England). Each dispensing aid was tested and evaluated by the committee, with an emphasis on ergonomics, reliable application, and patient safety. As the Opticare Arthro 5 and Opticare Arthro 10 seemed to be more challenging to use and could fit only small, rounded bottles, the Autodrop and the Opticare aids were approved by the committee for use in this study. The study was registered in the NIH clinicaltrials.gov website: https://clinicaltrials.gov/ct2/show/NCT03417453 (accessed on 21 November 2021).

For use, both devices are placed on the orbit. Figure 1 demonstrates the devices evaluated in the study. The Opticare device is assembled around the eye drop bottle. To dispense eye drops, the user must apply pressure on the device itself and not on the bottle. The Autodrop is used by inserting the tip of the bottle inside the device. When the user places the device onto the orbital rim, there is a small aperture that the user can look through, allowing a direct view for aiming.

Subject inclusion criteria were: age over 18 years, no previous regular eye drop use (sporadic use such as treatment for ocular infection or inflammation was allowed), and no motor disability that could negatively affect the ability to self-instill eye drops, as was defined in prior studies (tremor, inability to raise the arm or extend neck) [6,10].

For each subject, the drops instillation order was randomized using Randomization.com (http://www.randomization.com, accessed on 21 November 2021) and the instillation sequences were recorded in sealed envelopes labelled with the patient number. Each subject underwent a trial of all three instillation methods (standard eye drop bottle, the Opticare aid, the Autodrop aid) in a randomized order, obtained from the envelope. Artificial tears (Lyteers, Dr. Fischer, active ingredients: Hydroxyethylcellulose 0.19%) were used.

Subjects were instructed on how to use each delivery device and were allowed two trials. After the trials, subjects were instructed to administer exactly one drop into their right eye. The investigator observed and recorded the results according to a pre-written format. The number of delivered drops using the devices was assessed as follows: 1. If the subject did not fit the device properly then the investigator saw the number of drops directly; 2. If the device was used correctly and covered the eye, then the investigator assessed the amount of fluid accumulated in the eye and on the cheek/eyelid. This was evaluated using a subjective report on whether drops were instilled on the cheek/eyelid (yes/no) and by counting the drops by the study personal who visually assessed the instillation; 3. If no accumulation of fluid in the eye was seen and the subject reported drop instillation, the investigator concluded that only one drop was instilled. Otherwise, the investigator concluded that no drops at all were expelled. After the use of each device, subjects completed a questionnaire (Table 1) concerning the ease of use and the efficiency of each device. The questionnaire answers had a categorial scale (1–10, 1 = worst experience, 10 = best experience) or a binary scale (yes or no). Prior to the study, we reviewed the literature and constructed the questionnaire based on previous reports on eye drop aids. The questionnaire was based on surveys performed by Gomes et al. [11], Parkarri et al. [6] and Salyani et al. [4] regarding administration and satisfaction of patients using eye drop aids and was translated to Hebrew. The different questionnaires were evaluated, and relevant variables to our study were used.

### 2.2. Sample Size

To estimate the minimum sample size, we used the difference in the number of drops dispensed in two recent studies. The first was a study of 23 patients that compared the instillation of eye drops with and without an assisting delivery device [11]. The second study was performed on 40 subjects and assessed the accuracy, usage and contamination rates of an upright eye drops bottle compared to a conventional bottle [20]. Based on these studies, the mean difference in drops dispensed was set as 0.6 ± 1.1. For a paired model, a sample size of 26 patients was needed to detect a significant difference between groups, with a significance level of 0.05 and a power of 80%.

### 2.3. Statistical Analysis

Clinical parameters distribution was tested for normality by the Shapiro–Wilk test. Independent and paired t-tests were conducted for continuous variables with a normal distribution. The Wilcoxon signed-rank test and the Mann–Whitney U test were used for variables with a non-normal distribution. The McNemar test for paired proportions was used. Statistical analysis was performed using the SPSS software (version 25.0). A *p*-value < 0.05 was considered statistically significant. Data are presented as mean ± standard deviation (SD).

## 3. Results

There were 26 subjects (54% Male) were enrolled in the study with a mean age of 51.4 ± 21.6 years.

The objective and subjective measurements of the two aids and the conventional bottle for eye drop instillation are shown in Table 2. More than 90% of the subjects were able to assemble the eye drops bottle into the aid; however, the patients found the bottle was more easily assembled in the Autodrop than in the Opticare (mean difference of −0.9, 95% CI: −1.6802 to −0.1659, *p* = 0.02, Table 2). As compared to the conventional bottle, more drops were instilled when the Autodrop was used (median of 2 drops using the Autodrop and 1 drop using the conventional bottle, *p* = 0.05). In 35–38% of the patients, the eye drops fell either on the eyelids or on the cheek when using the two aids or the conventional bottle.

Regarding the risk of contamination (defined as touch of the bottle tip with the eyelids or an eye structure), no subjects using either the Autodrop or the Opticare had the bottle tip touch their eye, while 46% of subjects had contact with their canthus, eyelid, cornea or conjunctiva with the tip when using the conventional bottle without an aid (*p* = 0.0005). More subjects preferred Autodrop over Opticare (42% vs. 27%), but this was not significant (Table 2).

In addition to answering the questionnaire, subjects were invited to give free comments regarding their experience. While using the Autodrop, some reported that their gripping hand blocked the viewing aperture, and others had difficulty aiming the drops into their eyes and required multiple readjustments after failed attempts. Before using the Opticare, some subjects were intimidated by the look and size of the device. Some subjects who had never been able to independently instill eye drops before were able to do so using the two aids.

## 4. Discussion

In this study we compared both efficacy and safety of two drop aid devices and the conventional supplied bottle for independent eye drop instillation. The use of either drop aid completely prevented bottle tip contamination, which was frequently observed when the conventional bottle was used alone. When using the Autodrop, more drops were expelled in each application compared to the conventional bottle. Almost all subjects were able to assemble the eye drop bottle into both aids.

Contamination or trauma from the tip of an eye drop bottle is a major concern for patients taking topical treatment [21]. There have been reports of eye infections associated with contaminated bottles [22,23] and reports of trauma leading to conjunctival inflammation [24]. We found that 46% of subjects contaminated the bottle tip when using the conventional bottle. Dietlein et al. reported that 61% of glaucoma patients contaminated the bottle tip [10]. In another study, only 21.9% (using a 15 mL bottle) and 31% (using a 2.5 mL bottle) of patients with ocular diseases were able to instill a single drop without the tip of the bottle touching their eyes. This is despite the fact that 92.3% declared that they had no problems instilling the drops [18]. According to our results, bottle tip contamination was less frequent.

When using the Opticare, subjects needed to apply force onto the device and not to the eye drop bottle itself; thus, some subjects overestimated the force required to administer a single drop, leading to multiple drops being expelled from the bottle. Connor el at. [16] found that although the Opticare requires more pressure to expel eye drops, the required “key pinch” or “hand pinch” is easier to achieve than the “tip pinch” needed for conventional eye drop bottles [16,17]. As such, aids can enable eye drop self-administration in patients with motor difficulties who would be otherwise unable to do so. This was shown in a study that included subjects with rheumatoid arthritis and Sjögren’s syndrome [15]. The authors reported that half of all patients who used the conventional bottle had difficulty instilling eye drops, as opposed to only one-third who had difficulties using the Opticare. In addition, 13% of patients who needed the help of their spouse in eye drop instillation became independent with the Opticare [15].

Gupta et al. evaluated eye drop instillation in glaucoma patients and found that 50% of them instilled 2 drops or more (instead of 1 drop) [25]. They have speculated that excess medication instilled in the eye and drained by the lacrimal apparatus can increase absorption, and possibly the likelihood of unwanted systemic side effects. Medication wastage may also decrease a patient’s compliance, because the eye drops will run out faster, and more visits to the doctor or pharmacy for a repeat prescription will be needed. It also adds to the cost of treatment [9]. According to our study, Autodrop may lead to the instillation of multiple drops and wastage. However, this statement should be reserved as Autodrop was used after only two attempts, and more practice time with the device may reveal different results.

The participants in the current study were healthy people without arthritis, neurological disabilities or other diseases that might compromise drop instillation. Accordingly, our results may not be applicable to patients with systemic diseases that might compromise drop instillation. Future studies should include patients with physical limitations and also compare experienced versus nonexperienced patients in eye drop instillation. In addition, in this study, the participants had two attempts to use the eye drops aids before the experiment initiation. It is reasonable to assume that with continued use of the devices, the patients will better use them, with enhanced accuracy in administration. Our results can be utilized to assess the ease of acquiring the skill of using the eye drop aids but cannot be easily generalized for all future uses of the aids. Repeating the experiment either immediately or after a set amount of time and examining whether the results have changed would help make the results more generalizable. Based on previous studies [11,20], the mean difference in drops dispensed was set as 0.6 ± 1.1. A 0.6 drop difference, although significant statistically, is probably not significant clinically, which may make it difficult to interpret our results.

In conclusion, aids to administer eye drops such as Autodrop and Opticare were found to reduce the risk of bottle tip contamination. Assemble of the eye drops bottle into the aid was easy in both aids. These aids could have an advantage, especially in patients who had subconjunctival hemorrhage or corneal erosion from the bottle tip or had an eye infection that was attributed to the tip touching the eye.

## Figures and Tables

**Figure 1 jcm-10-05658-f001:**
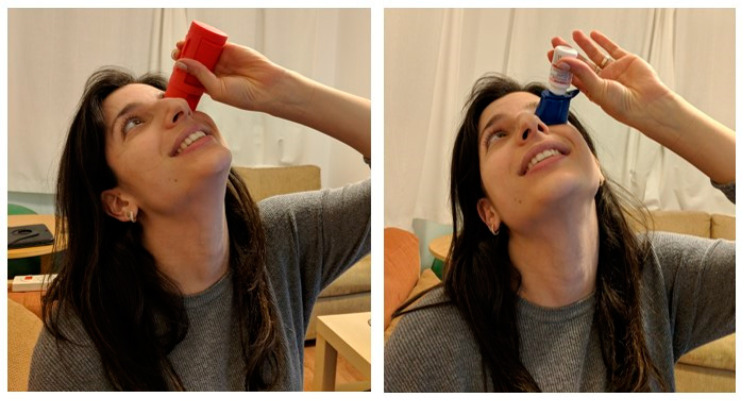
Devices evaluated in the study. (**Left**): Opticare, (**Right**): Autodrop.

**Table 1 jcm-10-05658-t001:** Patient’s questionnaire. Examinee number: _________. Please answer the questions below. Your response should be from a scale of 1–10 (10—best experience, easy to use; 1—worst experience, difficult to use).

	Standard Eye Drop Bottle	Opticare Aid (Blue)	Autodrop Aid (Dark Blue)
1. Ease of assembly of bottle into aid 1–10			
2. Ease of expelling drops 1–10			
3. Controlling number of drops discharged 1–10			
4. Controlling precise orientation (to the eye) 1–10			
5. General impression and satisfaction from the device 1–10			
6. Was there a wandering drop on your cheek or eyelid? Yes/no			
7. Willingness to use the device in the future Yes/no			
8. What is your favorite device? standard eyedrop bottle/Opticare aid/Autodrop aid			

**Table 2 jcm-10-05658-t002:** Objective and subjective assessment of the Autodrop, Opticare and conventional bottle by 26 subjects.

	Autodrop	Opticare	Conventional Bottle	*p*-Value
Subject was able to assemble bottle into aid	96%	92%	N/A	NS
Contamination of the bottle tip	0%	0%	46%	0.0005
Number of drops instilled, median (IQR)	2 (1–2)	1 (1–2)	1 (1–1)	0.05 for Autodrop- conventional comparison; NS for other comparisons
Drops instilled on cheek/eyelid (objective)	38%	35%	35%	NS
Ease of assembly of bottle into aid, mean ± SD	8.6 ± 1.1	7.7 ± 2.1	N/A	0.02
Ease of instilling drops, median (IQR)	8.5 (7–10)	8 (8–10)	8.5 (8–10)	NS
Ease to control the number of drops, median (IQR)	8.5 (8–10)	9 (7–10)	9 (8–10)	NS
Ease of aiming into the eye, median (IQR)	7 (4–9)	7.5 (7–9)	8 (6–9)	NS
Drops instilled on cheek/eyelid (subjective)	58%	50%	50%	NS
General impression and satisfaction with device, median (IQR)	7.5 (5–9)	8 (7–9)	8 (6–9)	NS
Willingness to use the device in the future	72%	72%	88%	NS
Favorite device	42%	27%	31%	NS

NS: nonsignificant; IQR: interquartile range; SD: standard deviation.

## Data Availability

The data presented in this study are available on request from the corresponding author. The data are not publicly available due to privacy.

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
