# Peer review of "Comparison of the Usability of Eye Drop Aids and the Conventional Bottle"

_jcm, 2021, doi:10.3390/jcm10235658_

Round 1
Reviewer 1 Report
Thank you for allowing me to review the article titled: ”Comparison of the efficacy and safety of eyedrop aids and the conventional bottle”. It is apparent that the authors have invested time and thought into the study.
Drug compliance is an important issue significantly influencing patients’ outcomes. This is even more true when dealing with eye drops, where technical difficulty is another obstacle that the patient needs to overcome.
The authors evaluated two eyedrop aids and a conventional bottle in a cross-over study of 23 participants. While the idea of assessing the utility of eyedrop aids is important, I think that some design flaws are present in the study at its current form limit the ability to gain significant insights from it.
My main concern with the study is that it does not assess the overall utility of the devices, but only a limited to assessing the ease of learning to use the eyedrop aids. This assessment still holds virtue, but the authors need to better describe it to the readership: The participants were introduced to two eyedrop aids, and after receiving an explanation, their ability to administer eyedrops properly was evaluated. I think it is reasonable to assume that the patients have never used the eyedrop aids before, however did use regular eyedrops previously. The study therefore evaluates the patients’ third attempt of using the two eyedrop aids vs their skill in using a regular eyedrop bottle. It is sensible to assume that with continued use of the devices the patients’ will use them better, leading to better accuracy in administration. Therefore, in my opinion, the results can be utilized to assess the ease of acquiring the skill of using the eyedrop aids but they cannot be generalized for all future uses of the aids. This makes statements about reducing waste, lowering side effects and increasing adherence far-reaching and un-supported.
This issue can be addressed by giving the participants more practice time with the devices (similar to ref 15 by Averns et al. in the submitted article), or comparing participants who are accustomed to using the devices. The findings can also be further solidified by repeating the experiment either immediately or after a set amount of time, examining whether the results changed. This would help make the results more generalizable.
I have listed my other concerns in a list-manner:
Introduction:
- The introduction mainly describes (17/26 lines) why eyedrop aids are needed. I think this part can be shortened, as the readership is familiar with this challenge.
I think that description of the eyedrop aids that are assessed and a more detailed overview of previous studies assessing eyedrop aids (do they increase compliance? Efficiency? Reduce waste?) is warranted, as I believe many of the readers are not familiar with the fine details of the topic.
Methods:
- Four devices were evaluated by the IRB, hospital’s medical engineer and an infectious disease specialist, but only two were approved for the study. Is there a reason for that?
- Was this trial registered with clinicaltrials.gov or a similar site? if not why?
- When the devices were placed properly, the investigator evaluated the amount of fluid in the eye and on the cheek / eyelid. It is not stated if this was a yes / no evaluation or an evaluation for the number of drops used. If the latter is the case, this adds a level of inaccuracy to the measurements.
- I do not possess the necessary statistical skills to assess the adequacy of the tests employed. However, I noticed that the Power calculations were based off different eyedrop aids (reference 11 assessed Xal-Ease; reference 17 assessed upright eyedrop bottle), that is possibly not equivocal to the two eyedrop aids assessed in the current study. Additionally, the power calculations were set for detecting a difference based of the means of these two studies, and not of a pre-determined clinical difference. Therefore the difference was set at 0.6 drops, which makes little sense clinically.
The study did show a statistically significant difference in the number of drops used, but a discussion about the clinical significance of the difference is lacking.
Results:
- Line 130: the statement should be changed to something along the line of ‘the patients found the autodrop to be easier to assemble’ since the question it is referring to is subjective
- Table 2: Contamination of bottle tip was not defined. Is it any touch with any eye structure? Was the tip cultured? Please define clearly.
- In Table 2 Ease of assembly in described by mean +/- SD, while other questions using 1-10 scale are described using median + IQR. Why the change?
- Describing the mean number of drops used will also help get a better understanding of the data as the IQR is similar for both eyedrop aids.
- Lines 146-154: The ‘free comment’ part of the questionnaire is interesting. But some of it is an explanation of the participants’ comments which should be reserved for the discussion section of the article (lines 150-152).
Line 153-154: This sentence is particularly interesting. Which device helped the participant to self-administer drops? Was it one or the other?
Discussion:
- Line 170-172: This is a very strong statement not supported by the data. According to the study bottle tip contamination was less frequent, which can possibly lead to less infection / trauma. The authors did not follow the patients to see if any developed an infection / trauma due to the eyedrop bottle touching the eye.
- Line 175-176: Awkward wording.
- Lines 182-188: The study shows that after 3 attempts, the autodrop user uses more drops on average. This cannot be generalized to say ‘leads to waste’ .
- Line 184: Gupta et al. [ref 25] found that 50% instilled two or more drops, not 24%.
- Line 184-186: While logical, neither reference 25 nor reference 26 support the sentence. Neither studies show increased absorption, or increased systemic or local side effects.
Again, thanks for allowing me to review the article. It is evident that the authors invested time and effort into a clear and well written study.
Reviewer 2 Report
Although the study is well designed and executed, I am afraid that there are some important points that should be considered: 1. The title (Comparison of the efficacy and safety of eyedrop aids and the conventional bottle) is not related to the results of the study. In my opinion, the authors try to test the best usability or simplicity of the eyedrop aids, but not their safety or efficacy. 2. Methods: 2.1 I would like to know if the questionnaire is validated, and if not, how are the authors decided which are the right questions to do?. This should be better explained in the article 2.2 Although the calculation of the sample is well explained, I think that it is low for this kind of study 3. Conclusions: The conclusion about the contamination of the bottles is not supported by the results. Although it is true that there is less contact of the bottle with the eye, additional tests are needed to conclude that there is more contamination.Author Response
Reviewer 2
Although the study is well designed and executed, I am afraid that there are some important points that should be considered:
- The title (Comparison of the efficacy and safety of eyedrop aids and the conventional bottle) is not related to the results of the study. In my opinion, the authors try to test the best usability or simplicity of the eyedrop aids, but not their safety or efficacy.
Response: We have changed the title accordingly: "Comparison of the usability of eyedrop aids and the conventional bottle." (pg. 1 line 2)
- Methods: 2.1 I would like to know if the questionnaire is validated, and if not, how are the authors decided which are the right questions to do?. This should be better explained in the article 2.2 Although the calculation of the sample is well explained, I think that it is low for this kind of study.
Response: 1. Prior to the study, we have reviewed the literature and constructed the questionnaire based on previous reports on eye drops aids. The questionnaire was based on surveys performed by Gomes et al. (reference 11), Parkarri et al. (reference 6) and Salyani et al. (reference 4), regarding administration and satisfaction of patients using eyedrop aids and was translated to Hebrew. The different questionnaires were evaluated and relevant variables to our study were used. We have now clarified this in the Methods (pg. 3 lines 115-120).
- The statistical analysis was planned and executed by an expert medical statistic personal (Dr. Nira Koren-Morag, Statistician and Epidemiologist from the Sackler School of Medicine, Tel Aviv University, Tel Aviv, Israel), which advised that a sample size of 26 patients in a test-retest design is needed to detect a significant difference between groups, with a significance level of 0.05 and a power of 80%.
- Conclusions: The conclusion about the contamination of the bottles is not supported by the results. Although it is true that there is less contact of the bottle with the eye, additional tests are needed to conclude that there is more contamination.
Response: Following this remark we defined contamination as every touch of the bottle tip with the eyelids or an eye structure (pg. 5 lines 159-160). This definition is widely used in studies in this era. In the conclusion of the article, we now write that: "In conclusion, aids to administer eyedrops such as Autodrop and Opticare were found to reduce the risk of bottle tip contamination." (pg. 7 lines 232-234).
We would like to thank again the reviewers for their time and effort in improving our manuscript.
Round 2
Reviewer 1 Report
Thank you for letting me participate in the review of the article. It is apparent that the authors invested time and thought into it.
The revised article addresses several of the issues raised in the first review.
These are my current reservations / remarks:
Introduction:
- The revised introduction still does not expand about the aids being evaluated. A short explanation would familiarize the readership with the devices which despite being readily available commercially, are not ubiquitous in use. The interested reader needs to acquaint herself with the devices from other resources.
Methods:
- The authors agree that setting the power calculations at 0.6 drops makes little sense clinically, but is significant statistically.
This is a weird way to design the study, that makes little sense. It makes interpreting any of the measured results very difficult.
Discussion:
- The authors revision to pg 6. Lines 194-197 is still unfounded in data. The sentence in its current form: "According to our results, bottle tip contamination was less frequent, which can lead to lesser risk of infection or trauma." still hints for less risk for infection or trauma although this is not supported. Keeping only the first of the sentence will be more accurate.
- The authors revision to pg. 7 line 214, although better, is still misleading. There is no supporting evidence for increased systemic absorption of medication due to inadvertently applying the drops to the cheek.
Overall the article is well written, as it was even before the authors’ revision.
Thank you again for allowing me to participate in the process.
Author Response
Reviewer 1
Thank you for letting me participate in the review of the article. It is apparent that the authors invested time and thought into it.
The revised article addresses several of the issues raised in the first review.
Response: Thank you for the kind words.
These are my current reservations / remarks:
Introduction:
The revised introduction still does not expand about the aids being evaluated. A short explanation would familiarize the readership with the devices which despite being readily available commercially, are not ubiquitous in use. The interested reader needs to acquaint herself with the devices from other resources.
Response: We have now expanded about both aids as the reviewer requested: "The Autodrop (Owen Mumford Ltd, England) and Opticare (Cameron Graham Limited, England) devices are two of the aids commercially available (Figure 1). Both are compatible with most eyedrop bottles. When using the Autodrop, the eyedrop bot-tle clips into place, while the body of the device prevents blinking by keeping the lower eyelid open. It has a small pinhole which directs eyesight upward and away from the descending drops. Patients apply pressure directly onto the bottle to expel eyedrops. When using the Opticare device, the eyedrop bottle is placed inside the device, ena-bling a better grip. The device is placed on the eye and the eyepiece holds the upper lid, which helps overcome the blinking reflex while administering eyedrops." (pg. 2 lines 61-69)
Methods:
The authors agree that setting the power calculations at 0.6 drops makes little sense clinically, but is significant statistically.
This is a weird way to design the study, that makes little sense. It makes interpreting any of the measured results very difficult.
Response: Following this remark we have stated in the discussion that: "Based on previous studies [11, 17], the mean difference in drops dispensed was set as 0.6±1.1. A 0.6 drops difference although significant statistically, is probably non-significance clinically, which may make it difficult to interpret our results." (pg. 7 lines 230-233)
Discussion:
The authors revision to pg 6. Lines 194-197 is still unfounded in data. The sentence in its current form: "According to our results, bottle tip contamination was less frequent, which can lead to lesser risk of infection or trauma." still hints for less risk for infection or trauma although this is not supported. Keeping only the first of the sentence will be more accurate.
Response: We have deleted the end of sentence, stating now that: "According to our results, bottle tip contamination was less frequent." (pg. 6 lines 194-196)
The authors revision to pg. 7 line 214, although better, is still misleading. There is no supporting evidence for increased systemic absorption of medication due to inadvertently applying the drops to the cheek.
Overall the article is well written, as it was even before the authors’ revision.
Thank you again for allowing me to participate in the process.
Response: The assumption was that applying to many drops into the eye (not on the cheek) may increase systemic absorption through the lacrimal system. This was raised by Gupta et al. (reference 25). Following the reviewer's remark, we have rewritten this statement: "Gupta et al. evaluated eye drop instillation in glaucoma patients and found that 50% of them instilled two drops or more (instead of one drop) [25]. They have speculated that excess medication instilled in the eye and drained by the lacrimal apparatus can increase absorption, and possibly the likelihood of unwanted systemic side effects." (pg. 6 lines 209-213) After that part we write about eyedrops wastage (drops applied to the cheek).
We would like to thank again the reviewers for the time and efforts to improve our manuscript.
